# Refractive plasma optics for relativistic laser beams

Omri Seemann[1,2] ✉, Yang Wan [1,2] ✉, Sheroy Tata[1], Eyal Kroupp [1] & Victor Malka [1]

The high intensities reached today by powerful lasers enable us to explore the interaction with matter in the relativistic regime, unveiling a fertile domain of modern science that is pushing far away the frontiers of plasma physics. In this context, refractive-plasma optics are being utilized in well established wave guiding schemes in laser plasma accelerators. However, their use for spatial phase control of the laser beam has never been successfully implemented, partly due to the complication in manufacturing such optics. We here demonstrate this concept which enables phase manipulation near the focus position, where the intensity is already relativistic. Offering such flexible control, high-intensity high-density interaction is becoming accessible, allowing for example, to produce multiple energetic electron beams with high pointing stability and reproducibility. Cancelling the refractive effect with adaptive mirrors at the far field confirms this concept and furthermore improves the coupling of the laser to the plasma in comparison to the null test case, with potential benefits in dense-target applications.

The numerous studies in laser–plasma interaction in the relativistic regime continuously contribute to the development of laser wakefield accelerators[1–5]. Laser wakefield relies on the strong laser ponderomotive force that efficiently expels electrons from its path, creating a co-propagating density perturbation. The electric field values associated with such electron plasma waves easily reach hundreds of GV/m[6,7], far beyond conventional accelerators. Yet, the propagation of an intense laser pulse in a dense gas target is the source of many nonlinear phenomena that alter the laser pulse properties. During such interaction, the self-modulation of the laser phase modifies the laser beam properties both spectrally (red and blue-shift), temporally (pulse compression or self-modulation), and spatially (filamentation, beam break up and self-focusing or defocusing)[8,9]. These changes in the laser pulse parameters are very sensitive to the gas density profile. For the purpose of optimizing the interaction, several techniques to tailor the plasma density profile have been explored: (i) Longitudinally, with mechanical gas flow obstacles to create a sharp shock front for controlling electrons injection[10] or with a density up-ramp in order to rephase the

acceleration process[11]. (ii) Transversely, plasma channels are used to extend the acceleration distance, formed using either an electric discharge[12,13] or a laser beam[14–18]. The latter method, which is the focus of this work, employs heating, also sometimes called a machining beam, that co-propagates with a main pulse. Through optical field ionization, the gas is transformed into plasma. Heated by the laser pulse, the plasma expands radially, producing at the interface with the surrounding gas a thin compressed layer and letting a lower-density rarefied region in the center of the plasma column. Because of the axial geometry of irradiation, it is difficult to accurately control the longitudinal density profile. Therefore, transverse optical shaping was conceived[19] showing advantages in triggering injection[20] and tuning the acceleration length[21]. Despite these successes and growing interest, an approach to longitudinally shaping targets that also enable plasma optics performances has yet to be established.

Here, we demonstrate the ability of transverse optical shaping to design suitable two-dimensional plasma optics in order to improve and control high-intensity, high-density interactions. For example, we

[1]Department of Physics of Complex Systems, Weizmann Institute of Science, Herzl 234, Rehovot 7610001, Israel. [2]These authors contributed equally: Omri Seemann, Yang Wan. ✉e-mail: omri.seemann@weizmann.ac.il; yang.wan@weizmann.ac.il

deliver multi-beams of relativistic electrons or, in the case of a single electron beam, increase the beam's energy and charge in the high-density regime that is normally susceptible to filamentation instability.

## Results and discussion

The experiment was carried out on HIGGINS dual 100TW laser system at the Weizmann Institute of Science. The two laser beams interact under vacuum conditions with a Nitrogen supersonic gas jet. The heater-laser spatial intensity profile was created by imaging the 100 fs, 800 nm, 30 mJ beam into a 2 mm × 8 μm line focus, arriving 14 ns prior to the 30 fs, 800 nm, 0.7 J, 16 μm FWHM main beam. The line focus intercepts part of the gas jet producing an evacuated plasma slab aligned along the propagation axis of the main laser pulse. The slab is then followed by an unaffected gas region with a peak atomic number density of approximately $7 \times 10^{18}$ cm$^{-3}$. To discern the plasma density profile through shadowgraphic measurements, we set up a second-harmonic probe laser pulse that normally propagates to the main beam. This probe beam, co-propagating with the heating beam, enables to image at different times of the plasma and its interface with the surrounding gas. A typical image revealing the density variations is presented in Fig. 1. To describe the interaction geometry, the $z$-axis is termed the longitudinal direction, $x$ is the horizontal transverse and $y$ is the vertical transverse.

The heating laser beam creates an evacuated plasma over its entire propagation length (i.e., $x$ direction) and over $\Delta_z \approx 2$ mm length and $\Delta_y \approx 100$ μm width. Since the beam's energy is deposited along $x$ and $z$ dimensions that are much larger than the transverse $y$, the plasma expansion is similar to a planar-point-source explosion, ideally exhibiting an electron density profile of $n_e(y,z) \propto y^{\frac{3}{2}}$[22]. This low-density evacuated plasma is surrounded by an expanding shock wave that compresses the gas into a thin layer of a few times the initial density. Analytically, in the center of expansion, the density reduces to naught. However, such a condition is not achieved in the experiment due to the finite heating source size and the effects of heat conductivity. To assess the density clearance at the center of expansion, the second-harmonic Thomson scattering of the main beam, which is proportional to the electron density, was imaged from the top view. The transverse integral of the scattering intensity image, normalized at the position of the unperturbed gas assuming the 5th ionization state, and averaged over several shots, is plotted in Fig. 2b. The resulting correlated electron density profile reaches the center of expansion, a value lower than 10% of the initial density. This measurement is subject to uncertainties

related to the longitudinal evolution of the main laser intensity. Therefore, we also show a comparison to OpenFOAM 2D hydrodynamic simulations (see "Methods" section).

When propagating through this lower-density plasma slab, the main beam's spatial phase is modified, affecting its interaction with the subsequent high-density non-disturbed region. To be effective as a plasma optic, the lower-density plasma requires strong enough variations of the refraction index $n(y, z)$ across the main beam's waist. The spatial phase can be assessed by integrating the optical path length multiplied by the vacuum laser wave number $k$: $\phi \approx k \int n(y,z)dz \approx -\frac{k}{2} \int \frac{n_e(y,z)}{n_c} dz$. Here, we have taken the first term of the plasma index of refraction, which is dependant on the electron density $n_e$, where $n_c$ is the critical density. Because the density varies in the $y$ direction (see Fig. 2d) but not in the x, the phase term gained in this target results in astigmatism. This can be expressed by the lateral spatial frequency ($k_y$): $\frac{k_y}{k} = \frac{1}{k} \frac{\partial \phi}{\partial y}$ and is calculated for different longitudinal positions of the target by integrating the simulated hydrodynamic 2D density map along the $z$-axis (see Fig. 2e). It is estimated that a few tens of milliradian phase shift over $10 \mu m$ transverse scale are achieved before the main laser is reaching the shock structure at $z = 0$.

Subsequently, in the high-density region, the main beam is observed to break up into filaments intense enough to drive their own wakefields and accelerate multiple relativistic electron beams, with a similar peak energy of $33 \pm 4.3$ MeV and charge per beam of $12 \pm 8$ pC. Furthermore, unexpected pointing stability and consistency are observed in nearly a hundred consecutive shots. A standard deviation of approximately 4.2 mrad and angular separation of 40 mrad is measured (see Supplementary Note 1 and "Methods" section). The occurring high-intensity, high-density interaction is sensitive to the vertical and longitudinal overlap between the heater and main laser beams; in the optimal conditions, 85% of the shots exhibit multiple electron bunches, with 70% of those being 3 beams aligned in the vertical direction.

The filaments occur through the process of Kerr instability arising from a non-linear index of refraction term ($n = n_0 + n_2 I$). The instability can be described by the maximum growing filament spatial frequency ($K_m = k \sqrt{\frac{2n_2 I}{n_0}}$) and a corresponding longitudinal growth rate ($g_m = \frac{K_m^2}{2k}$)[23–25]. In the case of relativistic laser and plasma, the non-linear index of refraction is evaluated based on the relativistic mass correction, which depends on the local laser amplitude: $\frac{n_2 I}{n_0} \approx \frac{n_e a^2}{4 n_c}$, where $a$ represents the laser normalized vector potential. We thus obtain a density-dependent expression for the filament frequency and growth rate ($K_m \approx ka \sqrt{\frac{n_e}{2n_c}}$, $g_m \approx ka^2 \frac{n_e}{4n_c}$). Based on this formula, in Fig. 2c, we plot the filament growth length (i.e., the inverse of the growth rate) as a function of the longitudinal position along the target, assuming $a \approx 1$ (similar to the experimental conditions). In the low-density part, the filamentation growth rate is low enough to prevent a strong modulation of the laser beam. However, for the later higher-density region, the filamentation process will break up the beam, resulting in multiple laser wakefields. Due to the gained astigmatism in the low-density region acting as a seed, this process can become stable, producing multiple electron bunches with the accurate beam pointing. It is of note that multi-beams of electrons have been produced before, using far-field laser optics situated prior to the principal focusing optic[26,27]. However, the difference here is that the additional phase shift is created at high intensity by the plasma itself.

In support of this scenario, 3D PIC simulations were carried out at the HZDR Hemera cluster using the code PIConGPU[28]. The simulations were initialized using an approximate density profile based on the hydrodynamic simulation but using fully ionized plasma with a peak electron number density of $4 \times 10^{19}$ cm$^{-3}$ and laser energy matching the experimental setup (further detail are described in the Methods section). The resulting 2D laser intensity transverse distributions (integral of volumetric distributions) are shown in Fig. 3, depicting the evolution

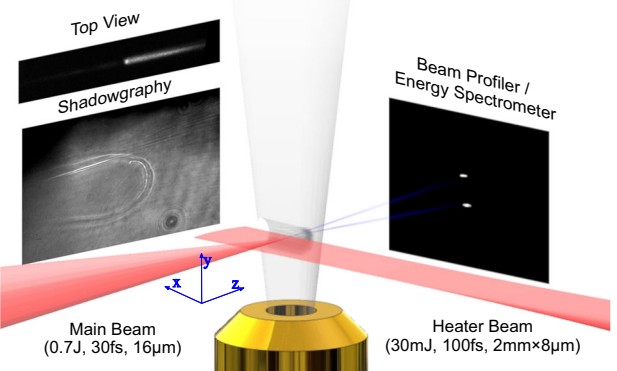

**Fig. 1 | Schematic of the experimental set-up.** In red (coming from the right), the heating laser beam that evacuates the plasma slab allows the main laser beam (coming from the left) to interact at high intensity with the denser part of the gas target. The plasma slab is probed using shadowgraphy techniques depicting the contour of the compressed region and using top-view imaging of second-harmonic, density-dependent Thomson scattering (shown on the left). On the right: a scintillator image showing two collimated electron beams, which is also used as an energy spectrometer by inserting a magnet in between.

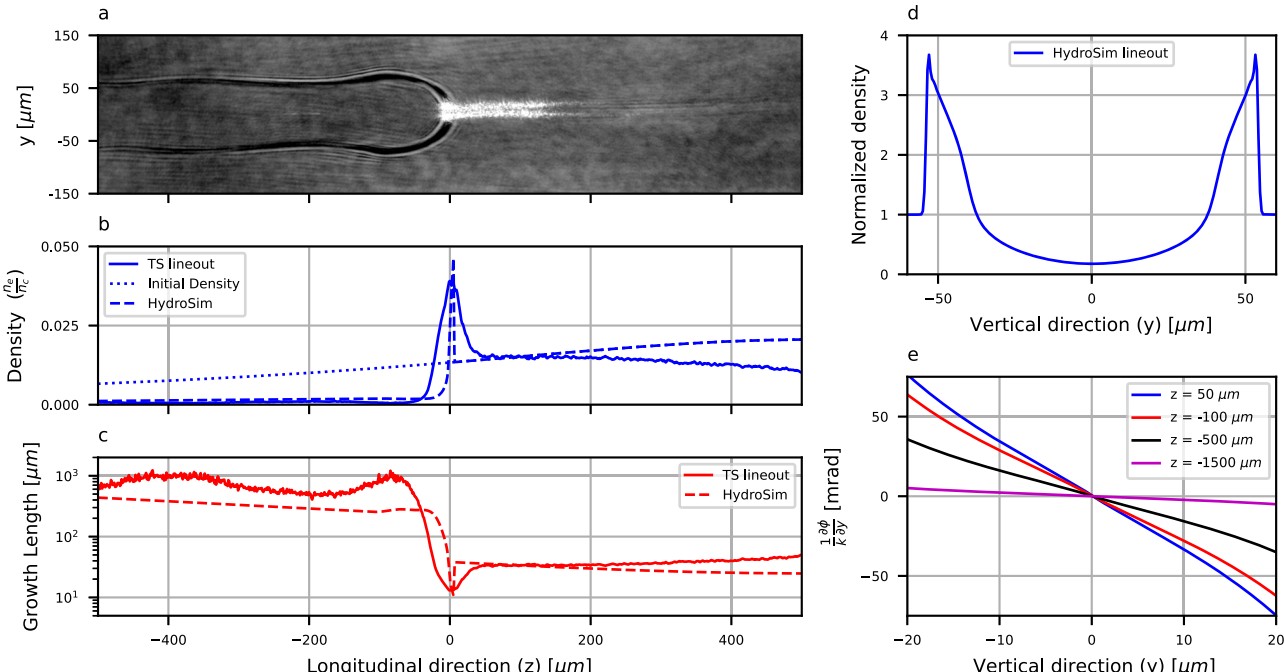

**Fig. 2 | The plasma density profile with and without the use of a heater laser beam. a** Shadowgraphy image (intensity in log scale), taken 14 ns after the heating beam. The main laser pulse propagates from left to right. In the low-density plasma slab on the left part of the image, the refracted probe beam reveals the shock boundary. The bright laser scattering to the right of the shock shows the formation of two intense filaments with angles matching the electron beams' separation. To enhance the contrast and laser scattering, the shot was taken with twice the atomic number density. **b** Longitudinal lineouts of the initial gas target density, the hydrodynamic simulation (HydroSim), and top view second-harmonic self-Thomson-scattering measurements normalized to initial density (TS). **c** The corresponding filamentation growth length calculations. **d** Vertical lineout of normalized density in the plasma slab based on hydrodynamic simulation. **e** Calculated gained spatial frequency inside the channel for several longitudinal positions of the target, carried by integrating the 2D density map of the hydrodynamic simulation.

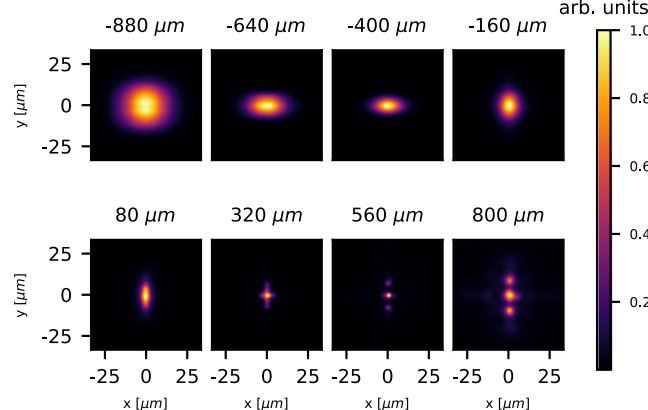

**Fig. 3 | The simulated main laser beam evolution in a plasma tailored by the heater laser.** Normalized laser intensity transverse distributions from a 3D PIC simulation at different longitudinal positions with respect to the channel's termination at $z = 0$ (annotated above each sub-figures). The accumulated astigmatic phase is evident in the evolution of the asymmetric focus changing from horizontal to vertical elongation. The laser beam is then split into three separated beams as it propagates inside the dense gas target.

inside the plasma for different longitudinal positions. The laser beam intensity near the focus exhibits an elongation that switches from horizontal to vertical directions as a consequence of the accumulation of the astigmatic phase. Importantly, also clearly visible at later times, the laser beam evolves into three distinct, intense filaments, which are responsible for the generation of the electron beams observed in the experiment.

To confirm experimentally the validity of the refractive plasma optic concept, we performed a test experiment by compensating the plasma-induced phase shift with an external phase-correcting adaptive-optic mirror. The amount of corrective term used was experimentally scanned, optimized, and measured by imaging the vacuum focal spot at different positions (see Supplementary Note 2.), with correction values comparable to the calculated spatial frequency obtained inside the channel (Fig. 2e). The results show that applying an astigmatic phase term totally suppresses the formation of the multiple beams. More interestingly, observed instead is a single beam with a significant increase in peak energy and charge ($64 \pm 9$ MeV, $44 \pm 13$ pC) shown in Fig. 4c, f. These electron beam parameters are also significantly better in comparison to those obtained without the heater beam ($41 \pm 3.2$ MeV, $17 \pm 8$ pC), as seen in Fig. 4b, e. In the latter case, before the laser beam reaches the denser central part of the gas target, a large fraction of it breaks up into several little filaments with a central one that contains still enough laser energy to drive a wakefield and produce the single electron beam. By evacuating the initial part of the plasma, the deleterious filamentation is mitigated, and higher laser intensity is reached in the high-density central region, improving the energy transfer from the laser to electron beams. The more pronounced mono-energetic peak is probably due to the shock structure at the interface between heated and nonheated regions that induces a localized electron injection[10,20] (Fig. 4). The sum effect is a non-negligible improvement in the laser-plasma accelerator performances for this relatively higher density scale target.

The generated multiple electron beams can be potentially applied for probing rapidly evolving plasma experiments[29], either for probing different longitudinal positions with a much wider field of view or the same position with a time delay between the beams created using a refocusing magnetic optic. In addition to the unique electron beams

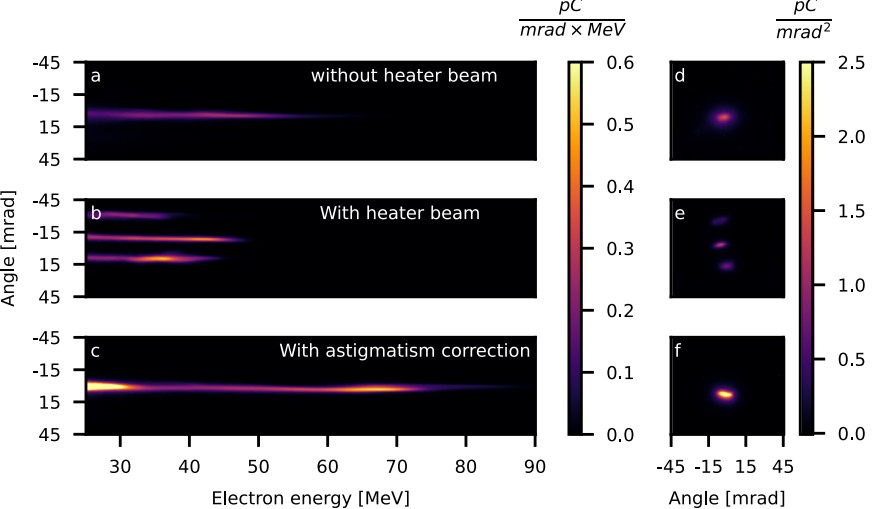

**Fig. 4 | Accelerated electron beam parameters.** Electron energy distribution with corresponding beam profiles measured for three cases showing the increase in beam parameters due to the combined plasma refractive optic and astigmatism correction: **a, d** Pure gas-jet target case. **b, e** Target with the heater beam. **c, f** Target with the heater beam and the external astigmatism correction.

and, more importantly, the reported plasma refractive optics concept opens two key prospects. First, it enables flexible laser beam manipulation in the 2D (longitudinal and transverse directions) through the control of the heater beam intensity profile, with further 3D manipulation possible through the use of holographic techniques. As evident from this experiment, this process is stable and robust enough to be used as near-field optical elements at the interaction region where the laser intensity is relativistic[30,31]. Secondly, in the context of near-critical-density gas targets, which require removing unwanted gas while keeping micron-scale density gradients[32,33], this concept is beneficial for preserving the laser phase and intensity during its propagation up to the designed interaction point inside the target.

As the cost of producing these relativistic plasma optics is negligible compared to those of conventional optics (the needed laser energy of the heating pulse represents a very small fraction of the total laser energy), they are promised to play an important role in the next future, especially for applications that required compact powerful laser and high repetition rate working conditions.

## Methods

### Main laser system
The Ti:Sa 800 nm laser system delivers two 30 fs laser pulses of about 1.5 J total energy each after compression. One of these serves as the main laser beam, which is first cut with a 40 mm diameter iris to improve spatial-phase-front quality and then focused using an off-axis parabola with a 500 mm focal length, resulting in an approximate Gaussian beam with an FWHM of 16 μm, 0.7 J on-target. An adaptive mirror is used to correct aberrations and, for this study, also to introduce the external astigmatic term. Images of the vacuum focal spots for different astigmatic conditions are shown in Supplementary Note 2.

### Target density fabrication and diagnostics
The second ultra-short laser pulse was tuned to 30 mJ and used to heat the gas target and produce a suitable plasma channel. This heating beam was imaged and focused using a 200mm focal length, 25 mm aperture cylindrical lens in order to create a line focus at the target position with vertical FWHM of approximately 8 μm. An intensity mask was imaged using a second cylindrical lens in the perpendicular orientation in order to create a controllable intensity profile with sharp boundaries at the target. The Nitrogen gas jet was released from a 1 to 2 mm diameter conical expanding nozzle connected through a pulsed valve with 4.5 Bars backing pressure reservoir. The nozzle exit was held 3 mm below the main beam focus. This setup provided an initial density profile approximated by a Lorentzian function with 1.4 mm FWHM and a peak atomic density of $n_{atom} \approx 7 \times 10^{18}$ cm$^{-3}$, which was verified with interferometric techniques. The jet center was placed approximately 500 μm after the termination of the heater beam (in the $z$-direction), and the main beam was focused within a Rayleigh length of the termination point.

The channel structure was imaged by using an additional second-harmonic (400 nm) ultra-short probe beam co-propagating with the machining beam. Intensity variations in the shadowgraphy image were due to changes in the refractive index induced by the plasma density profile. To prevent the bright glare of the main beam's Thomson side-scattering, an additional narrow band filter was used.

Probing the longitudinal density profiles was done by imaging the top-view second-harmonic self-Thomson-scattering. The corresponding amplitude is proportional to the electron density. The dependence on the main laser beam intensity is quadratic in the low-intensity regime and becomes closer to linear in the relativistic regime[34]. Integrating the intensity images in the transverse direction allows for diminishing of the dependence on laser intensity and provides an estimate of the electron longitudinal density profile.

### Electron beam measurements
A scintillator Lanex screen was positioned 40 cm after the target to measure the electron beam profile. Energy spectrum measurement was done by inserting a $1.6 \times 10^{-2}$ Tm magnet between the target and screen.

### Statistical analysis
The stable multiple electron beams were observed in various density conditions and relative positions of the heating and main beam foci, taken over several days with a few hundred shots each day. The standard deviations and absolute values of the beam energies and charges depicted in the manuscript were taken from the same day with the stated experimental conditions and smaller batches of consecutive shots: a batch of 29 shots for the 3 beam conditions (assuming all 3 beams come from the same charge and angular separation distributions), a batch of 10 shots for the corrected astigmatism conditions, and a batch of 14 shots for the null condition. The percentage depicting the stability of the phenomena was calculated based on approximately a hundred shots.

## Simulations

2D hydrodynamic simulations were carried out in order to evaluate the gained astigmatism and the instability growth rates and approximate the density for initialization of the 3D PIC code. The density in the experiment depends on dynamically changing ionization and dissociation states which were not taken into account. To approximate it, a mixture model was used where the gas was assumed to be diatomic nitrogen except inside a small box of 8 μm in height which was initialized with monoatomic nitrogen and 5 eV temperature to mimic the effects of the heater beam. The vertical position of the shock was matched to the shadowgraphy images in the experiment. Cell resolution was 0.5 μm.

3D PIC simulations were carried out using the PIConGPU code, which ran on the HZDR Hemera cluster. The spatial density profile was initialized with an approximate profile inside the expansion region as $\frac{n_e}{n_{e0}}(y,z) = T(y,z) \times L(z)$. Where $T$ is the transverse profile defined as $0.1 + 3R^{2.5} + 3R^{12}$ for $R < 1$ and one otherwise. $R = \frac{|y|}{50\,\mu m}$ for $z < 0$ and $\frac{\sqrt{y^2 + z^2}}{50\,\mu m}$ for $z > 0$ to capture the cylindrical interface near the termination of the heater beam. L is the longitudinal profile, i.e., a Lorenzian function. Input gas was fully ionized with a peak electron density of $4 \times 10^{19}$ cm$^{-3}$. The PIC simulation ran with a cell size of 130 nm in the transverse and 39 nm in the longitudinal direction, with approximately $4 \times 10^8$ cells on 12 V100 GPUs.

## Data availability

The data that support the findings of this study are available upon request to the corresponding authors.

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

## Acknowledgements

This work was supported by The Schwartz/Reisman Center for Intense Laser Physics, by a research grant from the Benoziyo Endowment Fund

for the Advancement of Science, by the Israel Science Foundation, Minerva, Wolfson Foundation, the Schilling Foundation, R. Lapon, Dita and Yehuda Bronicki, and by the Helmholtz association.

## Author contributions

O.S., Y. W., and V. M. have conceived and designed the experiment which was carried out by O.S., Y.W., and S.T. and the support of E.K.. O.S., and Y.W. analyzed the data and conducted the simulations. The article was written by O.S, Y.W., and V.M. with all authors discussing the results and commenting on the paper.

## Competing interests

The authors declare no competing interests.
