## [Peer Review File · Nature Communications]

Refractive plasma optics for relativistic laser beamsREVIEWER COMMENTS

Reviewer #1 (Remarks to the Author):

This manuscript describes the tailoring of laser pulses with refractive plasma optics resulting in either the acceleration of multiple electron beams by laser wake-field mechanism or by the acceleration of an enhanced (in terms of energy and emittance) single beam when using corrective adaptive optics mirror.

I think the work is very interesting and potentially deserving of being published on Nature Communications.

However, not working directly on this topic, I would like to ask the authors to expand a bit more about the potential applications of multiple LWFA beams and ask them in what sense this could represent a revolutionary find in the field.

Also I find remarkable that by using corrective optics is possible to substantially enhance the beam performance compared to a simple gas jet as well as to the non-corrected multiple beams. I would like to ask the authors how this result compares with other results in literature for optimized gas jet density profiles using different methods.

Finally the authors should add information in the methods section about the data statistical treatment including the number of shots conducted in the experiment. I saw that you included the percentage of shots exhibiting the multi- beam acceleration as well as errors in the measurement which makes me assume that you conducted the experiment with a large number of shots. Specifying it would add to the overall robustness of the results.

Best regards

Reviewer #2 (Remarks to the Author):

The paper "refractive optics for relativistic plasmas" describes interesting experimental results using a transverse ionizing beam to produce a structure in the target plasma that can control the properties of

the accelerated electrons. This paper is potentially publishable however I have several questions about the results and interpretation.

- 1) What is the density? In the Methods section it says experiments at $7 \times 10^{18} \text{ cm}^{-3}$ while the simulations are done with $7 \times 10^{19} \text{ cm}^{-3}$ (perhaps this is a typo). Can this not be measured with interferometry using the probe beam for comparison with experiments?
- 2) Why was pure nitrogen used? Would this not cause ionization defocusing?
- 3) Reference 20 is not given. Can the authors show the simulations showing decrease in density to 7% of initial density after 14 nsec? (as well as density measurements from the experiment).
- 4) why does a preionized plasma of 100 microns in extent (Figure 2) cause the laser beam to breakup into 3 components? The authors state that the transverse non-uniformity is significant but this looks quite uniform in Fig 2. What is the timing of the shadowgraphic measurement in Figure 2.

Reviewer #3 (Remarks to the Author):

The manuscript "Refractive plasma optics for relativistic laser beams" reports on an interesting experimental finding in the field of laser-wake field acceleration. By creating a cylindrical density boundary at the entrance face of a gas jet target by means of a heating laser beam, it is possible to reproducibly create three separate electron beams from one drive laser pulse. The authors attribute this behaviour to a astigmatic phase modulation imprinted on the beam by this cylindrical density boundary. Moreover, and perhaps a bit surprisingly, a pre-compensation of this astigmatic phase by a deformable mirror leads to a single electron beam with substantially better parameters than the reference case, i.e. without cylindrical boundary and pre-compensated phase. The experimental method is sound and the explanation given by the authors is plausible

The work is novel and potentially indicates a new route towards improved control of wakefield accelerators. Unfortunately, it is my firm belief that in its current form, the manuscript does not fulfill the criteria of Nature Communications on scientific stringency and general interest.

In order to generate sufficient general interest, the authors should elaborate on potential applications of their multiple electron beams, especially if these beams are not collinear. Their findings about the overall improvement of the electron beam quality are more compelling, but the manuscript lacks many important details, such as quantitative information about the longitudinal and transverse plasma profile with and without heater beam. Likewise, there is no quantitative information about the astigmatism caused by the cylindrical shock or the astigmatism introduced by the deformable mirrors to compensate for it. Furthermore, the authors present not even a simple back-of-the envelope calculation of the

expected astigmatism by the plasma and how this value would compare to the necessary compensation. Without these crucial information it is impossible for the reader to assess the validity of the author's explanation for their findings. Essentially, the author's explanation is nothing more than a hypothesis (albeit a potentially plausible one), and the lack of detail about the simulation results makes a quantitative comparison between simulations and experiment impossible.

In conclusion, sadly I cannot endorse publication of the manuscript in the present form, and the authors should add all available quantitative details about the experiment and simulation, as well as a quantitative comparison. Please find further comments in the attached edited pdf-file.

Response to Reviewer comments

General Response

We thank the editor and reviewers for considering our manuscript. We greatly appreciate their positive comments, constructive suggestions and questions. These remarks helped us to improve the discussion of our experimental observation and clarify the critical points within.

In our revision, we have thoroughly modified the content of the manuscript according to the reviewer suggestions, modified Figs. 1-3 and added more detailed information in the Methods and Supplementary based on the reviewers' comments. We have also double checked the analysis data and rerun the simulations to validate the various questions raised. We are convinced that the quality of this work and its relevance reach the high level of the Nature Communications publications.

In the following, we present a detailed response to all the comments (highlighted in blue) raised by the reviewers point-by-point, together with the relevant revisions.

Response to Reviewer 1

This manuscript describes the tailoring of laser pulses with refractive plasma optics resulting in either the acceleration of multiple electron beams by laser wake-field mechanism or by the acceleration of an enhanced (in terms of energy and emittance) single beam when using corrective adaptive optics mirror. I think the work is very interesting and potentially deserving of being published on Nature Communications. However, not working directly on this topic, I would like to ask the authors to expand a bit more about the potential applications of multiple LWFA beams and ask them in what sense this could represent a revolutionary find in the field.

We would like to thank the Reviewer's supportive comments. In fact, the key point of the paper is the novel optics made of structured plasmas in gas medium using a short low-energy laser pulse, and that are easy to produce even at high-repetition rate. This plasma based optic allows to manipulate the laser spatial phase at relativistic intensity (similar as the recent theoretical concept of holographic plasma lens [1, 2]). Here we introduced the astigmatism term, which could be extended to higher-order terms as long as a corresponding

plasma structure is created. Furthermore, this concept is also beneficial to near-critical-density gas targets, which can remove the undesired long pre-plasma region, significantly reducing the problematic laser filamentation, potentially enabling the production of high-charge particle and high-brightness x-ray beams. Therefore, we are convinced that this work demonstrate a revolutionary technique that can afford great benefits to the laser-plasma community.

The generation of multi beam electrons is an illustration offered by this concept. One important potential application of such beams is to probe rapidly evolving high-energy-density fields. For example, Ultra-short electron beams have been recently used for imaging highly nonlinear plasma waves [3] with an unprecedented precision revealing unique features of laser wakefield. Yet the field of view being limited, it requires several shot/event to have a full picture of the interaction. In case of warm dense plasma created with high energy and low repetition rate laser, this is a real limitation that is even more severe in cases where the initial conditions are changing from shot to shot. Spatially-separated multiple electron beams can provide a much wider field of view by probing at several target positions in the same shot (i.e. with exactly the same conditions), which is a big advantage compared with single electron beam. Alternatively, we can also probe the same position at multiple time delays by refocusing the electron beams with magnetic optics.

We have mentioned these in the conclusion section of the revised manuscript (lines 178-181).

Also I find remarkable that by using corrective optics is possible to substantially enhance the beam performance compared to a simple gas jet as well as to the non-corrected multiple beams. I would like to ask the authors how this result compares with other results in literature for optimized gas jet density profiles using different methods.

In the context of laser wakefield acceleration of electrons, the gas density in optimized target is much lower than the density used in our experiment. In those cases the laser is usually self guided or externally guided in a pre-formed plasma channel at a working plasma density of $10^{17} - 10^{18} \text{ cm}^{-3}$, which can result in a single electron beam with the energy from hundreds of MeV to several GeV level depending on the guiding conditions and laser energy. However, in our work, we did not try to optimize the "single" beam performances, but to show how this concept can significantly improve the interaction of the laser and its coupling with higher density gas targets.

Finally the authors should add information in the methods section about the data statistical treatment including the number of shots conducted in the experiment. I saw that you included the percentage of shots exhibiting the multi- beam acceleration as well as errors in the measurement which makes me assume that you conducted the experiment with a large number of shots. Specifying it would add to the overall robustness of the results.

This is very good suggestion. We have added a statistical section in the Methods (lines 241 - 251). The study is a fruit of a long (six weeks) experimental run, during which different parameters have been scanned, and with about

few hundreds shots every day during each of the week. We particularly made scans of the gas density, position of the machining beam focus and amount of astigmatism correction. As also presented in the Methods of the revised manuscript, the estimation of the appearance probability of multiple beams resulted from 83 consecutive shots on the same experimental condition. The standard deviations of the beam energy and charges were estimated based on the shots taken from the same day with similar conditions including a batch of 29 consecutive shots for the 3 beam condition, a batch of 10 consecutive shots for the corrected astigmatism conditions, and batch of 14 consecutive shots for the null test.

Response to Reviewer 2

The paper "refractive optics for relativistic plasmas" describes interesting experimental results using a transverse ionizing beam to produce a structure in the target plasma that can control the properties of the accelerated electrons. This paper is potentially publishable however I have several questions about the results and interpretation.

We thank for the Reviewer's positive comment and in the following we try to clarify all the points raised by the Reviewer and present the relevant modifications in the revised manuscript.

1) What is the density? In the Methods section it says experiments at $7 \times 10^{18} \text{cm}^{-3}$ while the simulations are done with $7 \times 10^{19} \text{cm}^{-3}$ (perhaps this is a typo).

The number $7 \times 10^{18} \text{cm}^{-3}$ stated in the Methods is the nitrogen atomic density measured in experiment. If assuming the gas is ionized to the 5th - 7th ionization state, this would give an equivalent electron density of about $3.5 - 5 \times 10^{19} \text{cm}^{-3}$. The latter stated electron density $7 \times 10^{19} \text{cm}^{-3}$ used for previous PIC simulations is indeed higher than the experimental one. We note that these 3D simulations were designed for a scaled down experiment carried out few months prior with slightly different conditions, which as well showed similar beam splitting results. In order to prevent confusion and better reflect the accurate experimental conditions, we have carried out a new set of 3D simulation study and the manuscript was revised accordingly. The modifications are listed as below,

I), The longitudinal and transverse plasma density profiles estimated from both experiments and hydro simulations, are added to the renewed Figure 2.

II), Figure 3 is changed according to the new 3D PIC simulations.

III), The simulation details are presented in the Methods section with modified electron density number of $4 \times 10^{19} \text{cm}^{-3}$ (line 268).

Can this not be measured with interferometry using the probe beam for comparison with experiments?

The measurement of the initial gas density profile (i.e. without heater beam) was performed with interferometry. With the heater beam, the created sharp gradients of the shock structure and the mixed gas-plasma states make

it almost impossible to reconstruct the actual density profile from the obtained phase information.

To infer the experimental density profile created by the heater beam, we instead speculate on the amplitude of the self emission of the (incoherent) Thomson scattering that is proportional to the electron density. By integrating the scattering light along the transverse axis, we obtained the longitudinal distribution of the plasma density, which has good agreement with the one from hydrodynamic simulations. In the revised manuscript, Figure 2b is renewed to present their comparison and we have also added the simulated transverse density profile in Figure 2d for better illustrating the phase modulation effect on the main laser pulse.

2) Why was pure nitrogen used?

In our experiment, we have tried two gas types: pure nitrogen and pure hydrogen. The phenomenon of multiple beams was first observed only with nitrogen. For hydrogen gas in comparison, it was observed from shadowgraphy that the formed channel vertical size is much larger due to the higher shock velocity, which introduces much less phase modulation to create sufficient astigmatic effect. Theoretically, using smaller delay between the heating laser and main laser could lead to smaller channel size that can trigger the astigmatic effect. In few hydrogen shots performed at higher densities which also enhance the astigmatic effect, we did see similar features. However we did not pursue these measurements enough to merit a discussion in the article.

Would this not cause ionization defocusing?

This is a very good point. For this range of density and laser parameters the refraction induced by ionization occurs for much longer propagation distance, and may explain the poor coupling of the laser and the electron beam in the case where we did not use the heating beam. To solely examine the ionization effect in the heated lower density region, we have performed small-scale Quasi-3D PIC simulations using the code FBPIC with ADK ionization model employed. The plasma density was set the same as that in the renewed Fig.2b. It is observed that for most of the laser propagation in the range of 1mm, the ionization state rapidly reaches 5 at two laser waists and only reaches slightly higher ionization states at the very center of the laser within few μm (see Fig. 1). Therefore, for most of the laser propagation in the heated low-density region, the ionization curve remains relatively flat in the lateral direction which should not create a strong defocusing effect.

Fig. 1 Laser intensity amplitude transverse profile in relativistic units (Blue). Ionization level from ADK in FBPIC (RZ) simulation (Orange). Most of the ionization induced gradient occur outside the region of the bulk of the laser energy, having a small effect on propagation.

3) Reference 20 is not given. Can the authors show the simulations showing decrease in density to 7% of initial density after 14 nsec? (as well as density measurements from the experiment).

We have added the missing reference and apologise for this. Regarding the density information, we have a detailed response to Q1 of the reviewer. Here we have to emphasize once again that to obtain the actual density profile is very challenging, especially for our complicated 2D structures shown in Figure 2a of the manuscript. On the experimental aspect, it is very difficult to reconstruct the density through interferometry or wave-front sensor due to the sharp gradient of the created shock structure and the transition between gas and plasma. Therefore, we deduced the density decrease using top-view Thomson-scattering measurements which are proportional to the electron density. On the simulation aspect, currently, no hydrodynamic simulation tools in the open source market, to our knowledge, contain the complete dissociation/ionization reaction models that fit to our experimental conditions. Therefore, the hydro simulations using OpenFoam from which the percentage was deduced have some uncertainties about the actual density profile after machined by the heater laser. We have tried our best to match the simulation to the experimental Thomson scattering data, by playing with parameters such as the viscosity and heating source size. Due to the uncertainty in the simulations, we have shifted the focus in the manuscript to the measured Thomson scattering intensity profile.

In the revised manuscript, We have modified the main text and Methods section to explain how the density profile has been estimated and to relate to

the measured scattering intensity (lines 83-92, 231-236, 253-261). The simulated density profiles and Thomson scattering deduced profile are now added to the renewed Figure 2b and 2d.

4) why does a preionized plasma of 100 microns in extent (Figure 2) cause the laser beam to breakup into 3 components?

This comment conducts us to amend the manuscript and add a more detailed explanation on the role of filamentation that appears to be very important in the context of this study. The breaking into 3 components occurs only when the laser reaches the higher-density un-machined region where the filamentation grows much faster. At the lower-density plasma of machined region, the transverse vertical phase is modified due the gradient of the refractive index of the created plasma slab, which induces the asymmetry in the laser wave-front phase leading to the the laser breakup only along one lateral axis.

In the revised manuscript, we have added to Figure 2c the filamentation growth rate curve. We have added Figure 2d and 2e to explain the astigmatic phase modulation on the main laser. Figure 3 shows the evolution of the laser transverse profile extracted from 3D PIC simulations depicting the increase of the laser astigmatism in the heated low-density plasma slab and the breakup of the beams in the high density region.

The authors state that the transverse non-uniformity is significant but this looks quite uniform in Fig 2.

The intensity modulation of the shadowgraphy image (Figure 2a) generally depends on second spatial derivative (Laplacian) of the refractive index or the plasma density. It is thus most sensitive to the shock interface surrounding the heated region, which has a density values of at least 30 times higher than the center region and far larger gradients (see Figure 2b and 2d). Therefore we cannot rely on the shadowgraphy image to conclude the uniformity in the low density region. Instead, based on either the analytical model of 1d explosion and the hydro simulations, it is reasonable that there is a transverse density gradient from the shock position to the center of the expanding plasma slab resulting from the rarefaction wave. The evaluated astigmatism based on the simulations are now shown in Figure 2e.

What is the timing of the shadowgraphic measurement in Figure 2.

The shadowgraphy was taken at a few tens of femtosecond later than the arrival of the main beam, and 14 ns later than the arrival of the heating beam. In the revised manuscript, we have added the timing detail in the caption of Figure 2.

Response to Reviewer 3

The manuscript "Refractive plasma optics for relativistic laser beams" reports on an interesting experimental finding in the field of laser-wake field acceleration. By creating a cylindrical density boundary at the entrance face of a gas jet target by means of a heating laser beam, it is possible to reproducibly create three separate electron beams from one drive laser pulse. The authors attribute

this behaviour to a astigmatic phase modulation imprinted on the beam by this cylindrical density boundary. Moreover, and perhaps a bit surprisingly, a pre-compensation of this astigmatic phase by a deformable mirror leads to a single electron beam with substantially better parameters than the reference case, i.e. without cylindrical boundary and pre-compensated phase. The experimental method is sound and the explanation given by the authors is plausible. The work is novel and potentially indicates a new route towards improved control of wakefield accelerators. Unfortunately, it is my firm belief that in its current form, the manuscript does not fulfill the criteria of Nature Communications on scientific stringency and general interest. In order to generate sufficient general interest, the authors should elaborate on potential applications of their multiple electron beams, especially if these beams are not collinear.

Firstly, we thank for the Reviewer's general positive evaluation on our work and detailed questions and suggestions we found benefiting. However, we would argue against the comment that our work provides insufficient general interest for Nature Communications Readers. The key significance of our work is that its demonstrate how the heater beam approach allows for a new concept to manipulate laser spatial phase at relativistic intensities, and that will largely benefit the high-intensity high-density laser-plasma interaction. Furthermore, as you have noted to, the increase in energy coupling is also important for optimizing the high density regime. Besides this point, the multiple electron beams can be used as an array of ultra-fast probes for diagnosing highly-transient intense plasma fields simultaneously at different positions. More detailed answer can be found in our response to the first remark of Reviewer 1 and also in the conclusion section of the revised manuscript.

Their findings about the overall improvement of the electron beam quality are more compelling, but the manuscript lacks many important details, such as quantitative information about the longitudinal and transverse plasma profile with and without heater beam.

Thanks for the Reviewer raising this point. In the revised manuscript, we have added in Figure 2 the longitudinal and transverse density profiles before and after the heating beam. Among them, the initial density profile is based on the interferometric measurements. The profile after machining is based on hydrodynamic simulations with parameters matching the experimental data we collected including the shock position based on shadowgraphy image and approximately to the density clearance at the center of the vertical channel from the top view image of the self-Thomson scattering. In the Methods, we have also added a section to explain how the simulation were set up (lines 252-261). More detailed answer about the density information can be found in our response to Q1 and Q3 of Reviewer 2.

Likewise, there is no quantitative information about the astigmatism caused by the cylindrical shock or the astigmatism introduced by the deformable mirrors to compensate for it.

Thanks for raising this point. Firstly, it is important to clarify that the astigmatism is not caused solely by the cylindrical boundary, but mostly by the

transverse density gradient of the long plasma slab (now shown in fig 2e). To confirm it, we have originally also conducted a separate 3D PIC simulation and replaced the cylindrical shock profile with a flat shock profile, where still clear astigmatism effect on the laser and its breakup into filaments were observed.

In the revised manuscript, we have now estimated the transverse accumulated astigmatism based on the hydrodynamic simulations and the results are plotted in the renewed Figure 2e. It is shown that before reaching the shock region, the accumulated phase shift at a transverse extent of around $10\ \mu\text{m}$ is around 30 mrad. In Fig.4 of the revised supplementary. We have also added the experimental measurements of the laser focal spot evolution in vacuum with and without the correction of the deformable mirror for comparison, where we can also estimate the introduced astigmatism to be comparable ($20\ \mu\text{m}$ lateral size over $500\ \mu\text{m}$ distance i.e. approximately 40 mrad). These values are also comparable to the angular separation between the electron beams.

Furthermore, the authors present not even a simple back-of-the-envelope calculation of the expected astigmatism by the plasma and how this value would compare to the necessary compensation. Without these crucial information it is impossible for the reader to assess the validity of the author's explanation for their findings. Essentially, the author's explanation is nothing more than a hypothesis (albeit a potentially plausible one), and the lack of detail about the simulation results makes a quantitative comparison between simulations and experiment impossible. In conclusion, sadly I cannot endorse publication of the manuscript in the present form, and the authors should add all available quantitative details about the experiment and simulation, as well as a quantitative comparison. Please find further comments in the attached edited pdf-file.

We agree with the Reviewer that it is beneficial to have more quantitative details on the analytical and simulation sides, though we have to emphasize that the comparison between the null case and the astigmatic case introduced by deformable mirror is essential to validate this hypothesis.

To improve this part, we have added a list of modifications to the revised version (some already mentioned in the response to the previous comment).

I), The renewed Figure 2 of the manuscript now contains the longitudinal density profiles before and after hydrodynamic evolution (based on 2d simulations), as well as the transverse profile after evolution and the resulting calculated astigmatic effect gained inside the channel at different longitudinal positions.

II), In Fig.4 of the Supplementary, we have added the focal-spot-evolution measurements in vacuum of the main beam, with and without adding the astigmatism from deformable mirror. The amounts of correction approximately matches the calculation based on the hydrodynamic simulations.

Following is our response to the specific comments given in the attached file

Figure 1, Indicate coordinates in figure

We have amended the figure.

Page 3, Where does the information about the thickness of the shock come from? Measurements? Simulations? Textbook knowledge?

Shock width as a function of mean free path is discussed in literature. However, as it is not pertinent to the topic of the paper, we have decided to remove the sentence.

Page 4, Please elaborate: What determines the critical power in the low density region? 16 and 20 TW are very close, with only a small deviation from the estimated band there would be effects also in the low-density region. Can you positively exclude a power above the critical power in any part of the low-density region?

This indeed was not precise enough. In fact, previous literature indicated that the laser pulse can reach self-focusing or self-guiding condition only when their power is several times of the critical power. However, the correlation is hard to quantify. In order to describe the physics more clearly, here we have modified the discussion from critical power to filamentation growth rate. As shown in the renewed Figure 2c, the filamentation grows much slower in the heated low-density region, such that it is negligible compared with the high-density region after the shock.

We have amended the discussion and figure 2 to present this argument (lines 119-139).

Page 4, I would like to understand the mechanism why the direction the filaments are spread out is oriented perpendicular to the heater beam (i.e. in y-dimension). From a beam hitting a cylindrical density step one would expect a cylindrical defocusing, producing a continuous line instead of three spots.

The astigmatism is created by the vertical transverse gradient of the heated low-density region, as we explain in the response to the third general remark raised by the Reviewer. The laser is then filamented in the high-density region developed from the seed produced by the astigmatism of the beam. This is why it would be spots and not a line. The filamentation occurs along the vertical axis is mainly due to the fact that the astigmatic laser beam after the focus is elongated along this direction (see Figure 3 of the revised manuscript). It is noted that the cylindrical boundary is not essential (and not the main contributor) for this process.

Page 5, What is the correct amount of astigmatism and how does it correspond to the figure you would expect from the effect of the cylindrical plasma lens? A reasonably quantitative back-of-the-envelope theoretical model should be included here.

We have added to the manuscript the expected astigmatism gained from hydrodynamic simulation in the renewed Figure 2e and the experimental corrected amount obtained from focal spot measurements in the Fig.4 of supplementary. They two have relatively good agreements. more details can be found in our response to the third and fourth remark raised by the Reviewer.

Page 6, This interpretation seems plausible, but not more. Without any details about the actual plasma profile the drive laser encounters it is very hard to substantiate your claim. You should provide this information

We agree with the Reviewer, and have added the detailed density information in the renewed Figure 2b and 2d.

Page 7, ??? 0.5 mm? 1x2 mm slit nozzle?

We are sorry for this confusing description. The gas nozzle is a conical expanding supersonic one. 1mm is the throat diameter and 2mm is the nozzle exit diameter. In the revised manuscript, we have corrected this part (lines 216-218).

Page 7, It is not at all clear how the density profile looks like. Please provide a lineout along the laser propagation axis. Moreover, as the heating beam is 1 mm wide, it may actually evacuate a significantly long portion of the whole gas jet, potentially leading to a much shorter acceleration region. Despite that, the electron energy is much higher than before. It is absolutely necessary to have a detailed understanding of the electron profile to understand what is really happening, but this information is completely missing.

We have added the longitudinal density profile in the renewed Figure 2b.

Regarding the electron beam energy differences presented in Figure 4 of the manuscript, it is true that the heated laser evacuates a large amount of the gas, leading to a shorter acceleration region in high-density region. However, the depletion length in the density above $3 \times 10^{19} \text{ cm}^{-3}$ is much less than 1 mm, therefore the acceleration length in our experimental condition is not the critical issue. Instead, it is the laser effective energy that matters. In the case without the heater laser, the main laser easily develops filaments before reaching its focus and exciting a nonlinear plasma wake to inject and accelerate electrons, which lose certain amounts of energy. In the case with the heater laser, by using deformable mirror to correct the introduced astigmatism, the main laser directly focuses on the shock with a much cleaner phase, where no significant filamentation occurs. The driven wakefield can therefore be much stronger compared to the former and the accelerated electrons get higher energy.

Page 7, A further suppression could potentially be achieved by making use of the fact that Thomson side-scattering is polarized along the laser axis.

Thanks for the suggestion. During experiments, we have also tried this by putting a polarizer in the shadowgraphy path however it did not show much improvements. The scattering polarization and angular distribution could be quite complex for relativistic lasers as there is electron motion also in the longitudinal direction and not only in the polarization direction.

Detailed modifications can be found in our attached manuscript with highlight changes, and we thank the Editor and Referees again for considering our work and believe that the presented modifications have indeed made the manuscript clearer and more accurate, and that it is now compliant with the high standards of Nature Communications publication.

Yours sincerely,

Omri Seemann, Yang Wan, Sheroy Tata, Eyal Kroupp, and Victor Malka.

References

- [1] Edwards, M., Munirov, V., Singh, A., Fasano, N., Kur, E., Lemos, N., Mikhailova, J., Wurtele, J., Michel, P.: Holographic plasma lenses. *Physical review letters* **128**(6), 065003 (2022)
- [2] Edwards, M.R., Michel, P.: Plasma transmission gratings for compression of high-intensity laser pulses. *Physical Review Applied* **18**(2), 024026 (2022)
- [3] Wan, Y., Seemann, O., Tata, S., Andriyash, I.A., Smartsev, S., Kroupp, E., Malka, V.: Direct observation of relativistic broken plasma waves. *Nature Physics* **18**(10), 1186–1190 (2022)

REVIEWER COMMENTS

Reviewer #1 (Remarks to the Author):

I've read the author's responses and the revised manuscript, including the questions and revisions asked by the other referees.

I think the manuscript is now suitable for publication in Nature Communications.

Reviewer #2 (Remarks to the Author):

The authors of "Refractive plasma optics for relativistic laser beams" have improved their paper and have provided more details about the experiment and the data analysis.

I find the work interesting however the explanation for the results is still somewhat lacking. The authors use the second harmonic of Thomson scattering to infer the density. However this is not proportional to the density but rather proportional to the product of the electron density and the square of the laser intensity which will make it difficult to use as a diagnostic for a plasma formed by a beam which is undergoing filamentation.

Also laser filamentation in a plasma is a beam propagation instability which evolves due to a non-linear growth of initial intensity modulations. Control of this instability into multiple beams for the stated application does not seem reasonable.

Although this work is interesting and publishable in some form the authors have not made the case that the result is significant enough for publication in Nature Communications.

Reviewer #3 (Remarks to the Author):

Since their first submission, the authors have considerably improved their manuscript and sufficiently addressed the largely common criticisms by the reviewers by substantial modifications. The manuscript

now is much better to read and it is easier to follow the physics involved. I now recommend publication of the manuscript as is.

Response to Reviewer comments

Dear Editor and Reviewers,

We would like to thank you for very supportive and constructive comments on the manuscript that helped us improve it, and especially for Reviewer 1 and 3 for recommending the manuscript for publication after the revision. We also find that even though Reviewer 2 found the work interesting, he/she now raised new concerns about the significance of our work. The significance of our work has already been discussed thoroughly in the previous round of the review and it have been fully recognized by the other reviewers. Apart from this concern, the reviewer brought up two questions about the data analysis to which we have a detailed reply in the following.

Response to Reviewer 2

I find the work interesting however the explanation for the results is still somewhat lacking. The authors use the second harmonic of Thomson scattering to infer the density. However this is not proportional to the density but rather proportional to the product of the electron density and the square of the laser intensity which will make it difficult to use as a diagnostic for a plasma formed by a beam which is undergoing filamentation.

Although the reviewer's statement is valid in the low intensity regime, We do not agree with it in our experimental conditions. Based on the nonlinear Thomson scattering theory [1] and experimental measurements [2], the second harmonic intensity increases roughly as the laser intensity I^2 for laser $a_0 < 1$, and saturates for $a_0 > 1$. In our case, $a_0 > 1$ and therefore the second harmonic signal has a weaker dependence on the laser intensity. As explained in the manuscript, we also perform hydrodynamic simulations to infer the density profile created by the heating laser beam that has been used in PIC simulations. By presenting together the intensity measurement and hydro-simulation results, we give a clear qualitative picture of the density profile, and leave to the judgment of the readers to conclude its ultimate accuracy. Again, we have to emphasize here that it is very challenging to find suitable tools for accurately diagnosing the density profile and the comparison shown in Fig.2b was the best approach we could achieve in current experimental conditions.

In the revised manuscript, we have clarified the dependence on the intensity of the Thomson scattering (lines 90-91, 233-239).

Also laser filamentation in a plasma is a beam propagation instability which evolves due to a non-linear growth of initial intensity modulations. Control of this instability into multiple beams for the stated application does not seem reasonable.

Here again we do not agree with the reviewer's statement. First, there are various applications of using controllable instability processes, such as Raman amplification, Optical parametric amplification, Free electron laser. Filamentation (similar to other instabilities) can also be controlled by reaching saturation and by controlling the seed. Self-focusing process which is the special case of filamentation, has been widely used in the laser wakefield acceleration that delivers at saturation stable electron beams. In our case, the instability is seeded by creating an initial phase modulation in low-density heated plasma. After reaching the high-density region, the instability develops and saturates to form multiple laser filaments driving their wakefields. The control of this filamentation process is verified by the stability of the multiple beams in nearly one hundred shots, which is also seen in the supplementary figures 1-2.

Yours sincerely,

Omri Seemann, Yang Wan, Sheroy Tata, Eyal Kroupp, and Victor Malka.

References

- [1] Esarey, E., Ride, S.K., Sprangle, P.: Nonlinear thomson scattering of intense laser pulses from beams and plasmas. *Physical Review E* **48**(4), 3003 (1993)
- [2] Chen, S.-y., Maksimchuk, A., Umstadter, D.: Experimental observation of relativistic nonlinear thomson scattering. *Nature* **396**(6712), 653–655 (1998)

Reviewers' comments:

Reviewer #2 (Remarks to the Author):

The paper "Refractive plasma optics for relativistic laser beams" describes experiments to use a plasma generated by a transversely propagating beam to perturb the propagation of an intense laser in order to generate multiple laser driven wakefields and therefore multiple electron beams. This is an interesting result but despite the response of the authors I do not consider this to be sufficiently important for publication in Nature Communications. The density measurement has large uncertainties and the explanation of the result as outlined in the paper and the previous responses (i.e., saturation of the filamentation instability) is not convincing.

The proposed phenomena also has limited potential applications when the same effect could be produced much more easily and with much better control using adaptive optics.

Response to Reviewer comments third round

Response to Reviewer 2

"The paper "Refractive plasma optics for relativistic laser beams" describes experiments to use a plasma generated by a transversely propagating beam to perturb the propagation of an intense laser in order to generate multiple laser driven wakefields and therefore multiple electron beams. This is an interesting result but despite the response of the authors I do not consider this to be sufficiently important for publication in Nature Communications."

We believe we made clear in various letters and responses to referees the importance of these results. They are not limited to the generation of multiple electron beams, which is just one consequence of the work. But include the creation of plasma refractive effect which can be shaped flexibly. We also would like to note recent papers published in high impact journals showing interest in this topic. Our results supporting the assertions about plasma refractive optics made by these papers, show for the first time the experimental implementation of a technique which could benefit the community at large without significant additional cost, and open avenues for new exploration in laser plasma interaction at high densities and for applications that require high repetition beams.

- "Optically controlled dense current structures driven by relativistic plasma aperture-induced diffraction" (Nature Physics 2016)
- "Plasma holograms for ultrahigh-intensity optics" (Nature Physics 2017)
- "Holographic plasma lenses" (PRL 2022)

"The density measurement has large uncertainties"

As replied in the second round of review process, the claim made about the proportionality is wrong for relativistic laser intensity (our case), for which the dependence of signal on laser intensity becomes much weaker. This is strongly supported by theoretical and experimental work (e.g. Nature 1999): "Experimental observation of relativistic nonlinear Thomson scattering".

Furthermore, the density shaping technique used in here can be given analytical density expectation alone, which we alluded to in the paper. We also show hydrodynamic simulation results supported by the Thomson scattering measurements which we believe together to be an accurate method pertinent

to this work. We also argue that the exact relation between intensity and density is irrelevant to the central topic of this work and is brought to support only the magnitude (not exact value) of the density reduction seen in simulations.

”and the explanation of the result as outlined in the paper and the previous responses (i.e., saturation of the filamentation instability) is not convincing.”

These arguments are scientifically false. It is known that the non-linear growth of instability is not contradictory of stability as soon as the saturation is reached, which is a well established theory applied for many applications such as laser self-focusing (which is a specific case of filamentation instability stated in our work), free electron lasing (beam-laser instability growth), Raman amplification (Raman instability growth) and a lot more to list. We also list in the manuscript itself several papers about stable multiple beam generation based on filamentation process due to external seeding.

In addition, we also showed in the current manuscript, 3D PIC simulations in support of this argument which the referee chose to ignore.

To summarize: there are experimental and simulation based evidence for the support of the claims made and their stability, and also historical evidence for the plausibility of the results. We do not understand what methods or arguments the referee 2 would value as convincing enough.

”The proposed phenomena also has limited potential applications when the same effect could be produced much more easily and with much better control using adaptive optics.”

The refractive effect by the plasma in this work appear in the near field rather than the far field. Suggesting the two are the same is ridiculous, ignoring fields such as holography.

Yours sincerely,

Omri Seemann, Yang Wan, Sheroy Tata, Eyal Kroupp, and Victor Malka.

REVIEWER COMMENTS

Reviewer #4 (Remarks to the Author):

The manuscript by Seemann et al. reports on a refractive plasma optics that can be used at relativistic laser intensities. An important advantage is that their method can be used at or close to the focus of very intense laser pulses. Before deciding whether the results are suitable for publication in Nature Communications, the authors should comment on the following points:

(1) It is well known that reshaping takes place in the case of an intense laser interacting with a dense gas target, which modifies the spatial, spectral and temporal properties of the laser pulse. The authors should discuss this in their manuscript. In particular, defocusing is expected to lower the peak intensity of the heating pulse. In this context, it seems surprising that the authors assume an ionization state of 5. The authors should explain how they come to this assumption and potentially add references.

(2) In the caption of Fig. 2 it says “Shadowgraphy image (intensity in log scale), taken 14ns after the heating beam, in a higher atomic number density case of $2 \times 10^{19} \text{ cm}^{-3}$.” It is unclear what ‘higher’ refers to.

(3) The authors state that a remarkable pointing stability is obtained and they say that a standard deviation of 4.2 mrad is measured. To set this in context, they should compare this to literature values in the main manuscript.

(4) On page 6 the authors state that they observed a significant increase in peak energy and charge. However, they compare these values only to their own results obtained without the plasma refractive optics. It would be important to compare these results to literature values to judge whether the results lead to a substantial improvement.

Response to Reviewer comments

Response to Reviewer 4

We thank the referee for considering our manuscript and the effort put into reviewing our work. On blue are his/her comments, which we reply to point by point. In red are modifications we added to the manuscript in agreement with the comments.

The manuscript by Seemann et al. reports on a refractive plasma optics that can be used at relativistic laser intensities. An important advantage is that their method can be used at or close to the focus of very intense laser pulses. Before deciding whether the results are suitable for publication in Nature Communications, the authors should comment on the following points:

(1) It is well known that reshaping takes place in the case of an intense laser interacting with a dense gas target, which modifies the spatial, spectral and temporal properties of the laser pulse. The authors should discuss this in their manuscript.

We agree with the referee on this pertinent comment and add as he/she requested a short discussion in the Introduction section on the important non linear effects that are produced when an intense laser pulse is interacting with a dense target : Yet, the propagation of an intense laser pulse in a dense gas target is the source of many nonlinear phenomena that alter the laser pulse properties. During such interaction, the self-modulation of the laser phase modifies the laser beam properties both spectrally (red and blueshift), temporally (pulse compression or self-modulation) and spatially (filamentation, beam break up or self-focusing or defocusing) [8,9]. These changes of the laser pulse parameters are very sensitive on the gas density profile. For this purpose several techniques to tailor the plasma density profile have been explored that allow to optimize the interaction... In addition to the methods that have been demonstrated these last decades and mentioned in the manuscript, we have demonstrated that the use of tailored plasma is an efficient way to control high-order spatial properties of relativistic laser beam.

In particular, defocusing is expected to lower the peak intensity of the heating pulse. In this context, it seems surprising that the authors assume an ionization state of 5. The authors should explain how they come to this assumption and potentially add references.

The referred sentence relates to the ionization state due to the main beam, and not due to the heater beam.

This question was asked by previous referees as well. We gave this reply:

Indeed, refraction induced by ionization occurs and plays a role when, for the given laser and gas density parameters, the gas is long enough. This could also play a role in the poor laser-gas coupling measured when no heating pulse is used. To solely examine the ionization effect in the heated lower density region, we have performed small-scale Quasi-3D PIC simulations using the code FBPIC with ADK ionization model. The plasma density was set the same as that in the renewed Fig.2b. It is observed that for most of the laser propagation in the range of 1mm, the ionization state rapidly reaches 5 at two laser waists and only reaches slightly higher ionization states at the very center of the laser within few μm (see Fig. 1). Therefore, for most of the laser propagation in the heated low-density region, the ionization curve remains relatively flat in the lateral direction which should not create a strong defocusing effect.

Fig. 1 Laser intensity amplitude transverse profile in relativistic units (Blue). Ionization level from ADK in FBPIC (RZ) simulation (Orange). Most of the ionization induced gradient occur outside the region of the bulk of the laser energy, having a small effect on propagation.

(2) In the caption of Fig. 2 it says "Shadowgraphy image (intensity in log scale), taken 14ns after the heating beam, in a higher atomic number density case of $2 \times 10^{19} \text{cm}^{-3}$." It is unclear what 'higher' refers to.

We thank the referee for this remark and indeed found the sentence confusing. "Higher" refers to a shot taken at twice the density of the results reported for the electron acceleration. This showed better contrast and a nice scattering of the filamented beam. We have now clarified this point in the figure caption.

(3) The authors state that a remarkable pointing stability is obtained and they say that a standard deviation of 4.2 mrad is measured. To set this in context, they should compare this to literature values in the main manuscript.

This "remarkable" was referring here to the idea that filamentation instability can be made stable if seeded properly. We agree that the choice of

adjective "remarkable" is not suitable and changed it by "unexpected". It is counter-intuitive that this instability can be stably seeded inside the target by the plasma itself.

(4) On page 6 the authors state that they observed a significant increase in peak energy and charge. However, they compare these values only to their own results obtained without the plasma refractive optics. It would be important to compare these results to literature values to judge whether the results lead to a substantial improvement.

We understand the concerns of the referee regarding the beam energy and charge parameters that are reaching (with similar laser pulse parameters) better performances than ours. It is important to mention that the goal of this article is not on optimizing the electron beam parameters which are affected by many variables such as target density and length, laser focusing optics, focus position and laser pulse duration, as it is likely impossible to reach them at few 10^{19}cm^{-3} densities compared to few $10^{17-18} \text{cm}^{-3}$. In the case of the higher densities, the literature on electron acceleration is characterized by significant variability [1–4], with electron beam distribution changing from monoenergetic to continuous spectra and electron energies ranging in 20 – 150 MeV for similar densities and laser energies. To isolate the heater beam effect alone, it was therefore essential to conduct our comparative analysis keeping the same gas target and laser parameters. This allowed us to draw meaningful conclusions about the benefit of the heater beam on the laser-plasma coupling.

Yours sincerely,

Omri Seemann, Yang Wan, Sheroy Tata, Eyal Kroupp, and Victor Malka.

References

- [1] Malka, V., Faure, J., Glinec, Y., Pukhov, A., Rousseau, J.-P.: Monoenergetic electron beam optimization in the bubble regime. *Physics of plasmas* **12**(5), 056702 (2005)
- [2] Hidding, B., Amthor, K.-U., Liesfeld, B., Schwoerer, H., Karsch, S., Geissler, M., Veisz, L., Schmid, K., Gallacher, J., Jamison, S., *et al.*: Generation of quasimonoenergetic electron bunches with 80-fs laser pulses. *Physical Review Letters* **96**(10), 105004 (2006)
- [3] Mangles, S.P., Thomas, A.G.R., Kaluza, M., Lundh, O., Lindau, F., Persson, A., Tsung, F., Najmudin, Z., Mori, W.B., Wahlström, C.-G., *et al.*: Laser-wakefield acceleration of monoenergetic electron beams in the first plasma-wave period. *Physical review letters* **96**(21), 215001 (2006)
- [4] Banerjee, S., Kalmykov, S.Y., Powers, N.D., Golovin, G., Ramanathan, V., Cunningham, N.J., Brown, K.J., Chen, S., Ghebregziabher, I., Shadwick, B.A., *et al.*: Stable, tunable, quasimonoenergetic electron beams produced in a laser wakefield near the threshold for self-injection. *Physical Review*

REVIEWERS' COMMENTS

Reviewer #4 (Remarks to the Author):

The authors have satisfactorily responded to my comments, and I can now recommend the manuscript for publication in Nature Communications.